# Stochastic Learning of Additive Second-Order Penalties with Applications to Fairness

## Abstract

Many notions of fairness may be expressed as linear constraints, and the resulting constrained objective is often optimized by transforming the problem into its Lagrangian dual with additive linear penalties. In non-convex settings, the resulting problem may be difficult to solve as the Lagrangian is not guaranteed to have a deterministic saddle-point equilibrium. In this paper, we propose to modify the linear penalties to second-order ones, and we argue that this results in a more practical training procedure in non-convex, large-data settings. For one, the use of second-order penalties allows training the penalized objective with a *fixed* value of the penalty coefficient, thus avoiding the instability and potential lack of convergence associated with two-player min-max games. Secondly, we derive a method for efficiently computing the gradients associated with the second-order penalties in stochastic mini-batch settings. Our resulting algorithm performs well empirically, learning an appropriately fair classifier on a number of standard benchmarks.

## 1 Introduction

Machine learning systems are becoming increasingly prevalent in real-world applications, consequently affecting the decisions that determine a person's life and future, such as playing a role in parole conditions (Angwin et al., 2016), loan applications (Guegan & Hassani, 2018), and airport screening (Guimaraes & Tofighi, 2018). Recent work has shown that such machine learning models often have biases which can unfairly disadvantage certain groups. For example, learned word embeddings exhibit gender-specific biases in what should be gender neutral words (Bolukbasi et al., 2016). In another case, a machine learning model's predictions regarding convict recidivism were found to be unfairly biased against African-Americans (Angwin et al., 2016). While it may seem at first that simply ignoring the features corresponding to these protected traits when training can alleviate this, previous work (Pedreshi et al., 2008) has shown that enforcing such blindness is largely ineffective due to redundant encodings in the data. In other words, while the learning algorithm used may not be biased, the data can be inherently biased in complex ways, and this leads to models which perpetuate these undesirable biases.

Research into the challenging problem of machine learning fairness is therefore of great interest. To better specify this problem, previous work has elaborated on precise notions of fairness, such as demographic parity (Dwork et al., 2012), equal opportunity (Hardt et al., 2016), etc. These notions can often be expressed mathematically as a linear constraint on the output of a machine learning model, taken in expectation over the entire data distribution. Accordingly, a number of recent works have proposed to incorporate fairness during training by expressing the objective as a constrained optimization problem (Zafar et al., 2015; Goh et al., 2016). If the original objective is convex, the addition of linear constraints results in a problem which may be readily solved by Lagrangian methods.

However, modern machine learning models are often not in a convex form. Indeed, the success of deep neural networks over the past decade makes it clear that the most well-performing models are often highly non-convex and optimized via stochastic gradient methods over large amounts of data (Szegedy et al., 2017; Wu et al., 2016). It is unfortunate that much of the existing work on fairness in machine learning has provided methods which are either focused on the convex, small data-set setting (Zafar et al., 2015; Goh et al., 2016), or otherwise require sophisticated and complex training methods (Cotter et al., 2018b).

In this paper, we present a general method for imposing fairness conditions during training, such that it is practical in non-convex, large data settings. We take inspiration from the standard Lagrangian method of augmenting the original loss with linear penalties. In non-convex settings, this dual objective must be optimized with respect to both model parameters and penalty coefficients concurrently, and in general is not guaranteed to converge to a deterministic equilibrium.

We propose to re-express the linear penalties associated with common fairness criteria as second-order penalties. Second-order penalties are especially beneficial in non-convex settings, as they may be optimized using a *fixed* non-negative value for the penalty coefficient $\lambda$. When $\lambda \to 0$ the optimization corresponds to an unconstrained objective, while as $\lambda \to \infty$, the problem approaches that of a hard equality constraint. This allows us to avoid sophisticated optimization methods for potentially non-convergent two-player games. Instead, we only need to choose a fixed value for the penalty coefficient, which may be easily determined via standard hyperparameter optimization methods, such as cross-validation. As an additional benefit, by choosing the penalty coefficient on a separate validation set, we can improve generalization performance.

Second-order penalties, however, potentially introduce a new problem: By squaring an expectation over the entire data distribution, the resulting penalized loss is no longer an expectation of loss functions on individual data points sampled from the distribution, and therefore not readily approachable by stochastic gradient methods. We solve this by presenting an equivalent form of the second-order penalty as an expectation of individual loss functions on *pairs* of independently sampled data points.

Our resulting algorithm is thus not only more practical to optimize in non-convex settings, using a fixed value for the penalty coefficient, but is also easily optimized in large-data settings via standard stochastic gradient descent. We evaluate the performance of our algorithm in a number of different settings. In each setting, our algorithm is able to adequately optimize the desired constraints, such as encouraging feature orthonormality in deep image autoencoders and imposing predictive fairness across protected data groups.

## 2 FAIRNESS AS LINEAR CONSTRAINTS

Consider a data domain $X$ and a probability measure $\mu$ constituting a data distribution $\mathcal{D} = (X, \mu)$. Let $H$ be a set of real-valued functions $f : X \to \mathbb{R}$ endowed with the inner-product,

$$\langle f, g \rangle = \int_X f(x)g(x)d\mu(x). \tag{1}$$

A machine learning model is a function $d \in H$. Given a data point $x \in X$ it provides a *score* $d(x)$. When the machine learning model is a predictive binary classifier, the range of $d$ is $[0, 1]$ and a score $d(x)$ corresponds to a machine learning model which on input $x$ returns 1 with probability $d(x)$ and returns 0 with probability $1 - d(x)$. Many notions of fairness may be written as linear constraints on $d$. That is, they may be expressed as $\langle d, c \rangle \in [C - \epsilon, C + \epsilon]$, where $c \in H$ is some other fixed function and $C \in \mathbb{R}, \epsilon \in \mathbb{R}_+$. We elaborate on a few popular notions below.

For simplicity, we restrict the text to refer to a domain $X$ with a single protected group $G \subset X$ whose predictions $d(x)$ we desire to be *fair* with respect to the distribution of predictions on the entire domain $X$. Nevertheless, all of our results apply to the fully general multi-group setting. We assume access to an indicator function $g(x) = 1[x \in G]$ and thus the proportion of data in $G$ is $Z_G = \langle 1, g \rangle$. We let $y : X \to \{0, 1\}$ be the true label function. Thus, the proportion of examples which are positive is $P_X = \langle 1, y \rangle$; the proportion which are positive and in $G$ is $P_G = \langle g, y \rangle$.

- **Demographic parity** (Dwork et al., 2012): A fair classifier $d$ should make positive predictions at the same rate on each group. This constraint may be expressed as $\langle d, c \rangle = 0$, where $c(x) = g(x)/Z_G - 1$.
- **Equal opportunity** (Hardt et al., 2016): A fair classifier $d$ should have equal true positive rates on each group. This constraint may be expressed as $\langle d, c \rangle = 0$, where $c(x) = g(x)y(x)/P_G - y(x)/P_X$.
- **Equalized odds** (Hardt et al., 2016): A fair classifier $d$ should have equal true positive and false positive rates on each group. In addition to the linear constraint associated with equal opportunity, this notion applies an additional constraint $\langle d, b \rangle = 0$, where $b(x) = g(x)(1 - y(x))/(Z_G - P_G) - (1 - y(x))/(1 - P_X)$.

- **Disparate impact** (Feldman et al., 2015): A fair classifier $d$ should have a rate of positive prediction on a group at least $p\%$ as high as the rate of positive prediction on another group. Traditionally, $p = 80$. Unlike the other notions of fairness, disparate impact *may not* be expressed as a linear constraint. Nevertheless, previous work (Zafar et al., 2015) has suggested approximating it as such; i.e. $\langle d, c \rangle \in [-\epsilon, \epsilon]$, where $c(x) = g(x)/Z_G - 1$.

## 3 NON-CONVEX OPTIMIZATION WITH SECOND-ORDER PENALTIES

Any linear constraint $\langle d, c \rangle \in [C - \epsilon, C + \epsilon]$ may be equivalently written as a quadratic constraint:

$$(\langle d, c \rangle - C)^2 \leq \epsilon^2. \tag{2}$$

We first discuss the known approaches based on the penalty-form Lagrangian with linear constraints and list their disadvantages. We then introduce the optimization based on our second-order penalties and how it avoids many of the issues associated with linear constraints.

### 3.1 CLASSICAL LAGRANGIAN WITH LINEAR PENALTIES

Suppose that we wish to minimize a loss $\ell(d)$ over classifiers $d$ subject to constraints $\langle d, c_k \rangle \in [C_k - \epsilon_k, C_k + \epsilon_k]$, $k = 1, \ldots, m$. The Lagrangian is formulated as follows

$$\mathcal{L}(d, \lambda) := \ell(d) + \sum_{k=1}^{m} \lambda_k (\langle d, c_k \rangle - C_k - \epsilon_k) + \sum_{k=1}^{m} \lambda_{k+m} (-\langle d, c_k \rangle + C_k - \epsilon_k).$$

Then, the original constrained optimization problem (the primal problem) would be $\min_d \max_{\lambda \geq 0} \mathcal{L}(d, \lambda)$ and the dual problem would be $\max_{\lambda \geq 0} \min_d \mathcal{L}(d, \lambda)$.

If our loss function is convex, then solving the dual problem would lead to the solution for the original problem; i.e., there is no duality gap. Unfortunately, this is not so in the non-convex case. Not only can the solution to the dual problem not yield an optimal feasible solution, there may not even exist a saddle point in the Lagrangian that we can converge to. Instead, in order to solve such a constrained optimization problem, one must consider it as a two-player game where one player chooses a classifier and the other player chooses a Lagrange multiplier. The final solution will then be a *mixed* equilibrium: in other words yields a *randomized* classifier. This line of work has received recent attention; e.g. Arora et al. (2012); Agarwal et al. (2018); Cotter et al. (2018b). To summarize:

- The Lagrangian when optimizing jointly over model parameters and multipliers may not converge to any saddle point, and even if converged, may not lead to the right solution. This requires us to resort to sophisticated procedures with many parameters and hyperparameters, which may be difficult to train.

- The Lagrangian approach with non-convex objectives leads to randomized classifiers, which may not be desirable in practice. Although work has been done to reduce the randomization (Cotter et al., 2018b), it is in general not possible to reduce the solution to a deterministic one while still being an equilibrium.

- Another weakness of the Lagrangian approach is that the constraints must be relaxed or approximated (e.g. hinge relaxation (Goh et al., 2016)) in order to make them convex and differentiable. The introduced slack necessitates hyperparameter tuning to encourage the optimization on approximated constraints to yield a classifier satisfying the original constraints.

### 3.2 OUR FORMULATION WITH SECOND-ORDER PENALTIES

We propose to optimize the following objective

$$\min_d \ell(d) + \lambda \sum_{k=1}^{m} (\langle d, c_k \rangle - C_k)^2, \tag{3}$$

where $\lambda \geq 0$ is a hyperparameter which decides the fairness-accuracy tradeoff. Note that $\lambda \to 0$ corresponds to the unconstrained objective, while $\lambda \to \infty$ corresponds to an optimization with hard constraints (i.e. $\epsilon = 0$). Any fixed value of $\lambda > 0$ will yield a solution between these two extremes.

Since the penalty coefficient $\lambda$ is fixed during training, it may be treated as an additional hyperparameter. Standard hyperparameter optimization methods may be used to choose $\lambda$ based on validation so that the final solution gives the desired fairness-accuracy trade-off. While we are not optimizing for the fairness metrics such as demographic parity or equalized odds directly, we will show that empirically our second-order penalties provide a reasonable proxy for many of the popular metrics. Additionally, while popular methods such as the aforementioned Lagrangian with linear penalties appear to directly optimize for the fairness metrics, in practice they require some sort of relaxation of these constraints to make the optimization feasible; thus in essence, such methods also optimize a proxy to the fairness metrics rather than the actual metrics desired and accordingly also require some amount of hyperparameter tuning.

It can now also be seen that this alternative way of solving for fairness-constrained classifiers overcomes many of the drawbacks listed earlier associated with methods based on the Lagrangian: For any fixed choice of $\lambda$, there will exist a solution to the optimization. Moreover, this solution will be deterministic. Next, by tuning $\lambda$ to directly satisfy the desired fairness metrics, we decouple much of the inherent difficulty of hyperparameter tuning in the Lagrangian approaches, which rely on the procedure itself to allow a certain amount of slack. Moreover, this decoupling encourages better generalization performance as there is less chance of overfitting to the training set compared to approaches which solve for the model parameters and Lagrange multipliers simultaneously on the same dataset.

### 3.3 Stochastic Optimization of Second-Order Penalties

At face value, it appears that the introduction of second-order penalties may complicate stochastic optimization of the objective. The quadratic penalty is a square of an expectation over the dataset. It is not possible to express such a penalty as an expectation of individual loss functions over the dataset. In this section, we show that despite this obstacle, it is in fact possible to express the second-order penalty as an expectation of individual loss functions over *pairs* of data points sampled from the dataset. This derivation is crucial for most modern machine learning applications, in which the data set is exceedingly large, at times presenting itself in an online form, and must be optimized using stochastic mini-batches.

The second-order penalty is of the form

$$\sum_{k=1}^{m}(\langle d, c_k\rangle - C_k)^2 = \sum_{k=1}^{m}\left(\int_X d(x)c_k(x)d\mu(x) - C_k\right)^2. \tag{4}$$

Since $\mu$ is a probability measure, we may re-write the integrals as

$$\sum_{k=1}^{m}\left(\int_X d(x)c_k(x) - C_k\ d\mu(x)\right)^2. \tag{5}$$

We may express each squared integral as a double integral:

$$\sum_{k=1}^{m}\int_X\int_X (d(w)c_k(w) - C_k)(d(x)c_k(x) - C_k)\ d\mu(w)d\mu(x). \tag{6}$$

Finally, we may express the sum of these double integrals as a double integral of a sum:

$$\int_X\int_X\sum_{k=1}^{m} (d(w)c_k(w) - C_k)(d(x)c_k(x) - C_k)\ d\mu(w)d\mu(x). \tag{7}$$

The gradients of this double integral with respect to parameters of $d$ may be approximated via Monte Carlo estimation, only requiring access to two independent samples $w, x$ from $\mathcal{D}$.

### 3.4 Algorithm

Algorithm 1 provides a pseudocode of our fairness-aware training algorithm. The machine learning model is parameterized as $d_\theta$, for parameter vector $\theta$. We also assume the training loss $\ell$ is an expectation of individual loss functions: $\ell(d_\theta) = \int_X \ell(d_\theta(x), y(x))\ d\mu(x)$.

---

**Algorithm 1** Fairness training with second-order penalties.

---

**Input:** Dataset $\mathcal{D}$, classification loss function $\ell$, constraints $\{c_k, C_k, \epsilon_k\}_{k=1}^m$, model parameterization $d_\theta$, learning rate $\eta$, number of training steps $N$, batch size $B$, hyperparameter $\lambda$.

Initialize $\theta^{(0)}$.
**for** $t = 0$ **to** $N - 1$ **do**
    Sample batch $\{(x_i, y_i)\}_{i=1}^B \sim \mathcal{D}$.
    Compute loss gradient $\nabla_\theta \hat{L} = \frac{1}{B} \sum_{i=1}^B \frac{\partial \ell(d_\theta(x_i), y_i)}{\partial \theta}\big|_{\theta = \theta^{(t)}}$.
    Compute penalty gradients $\nabla_\theta \hat{Q}_{ijk} = \frac{\partial}{\partial \theta}(d_\theta(x_i) c_k(x_i) - C_k)(d_\theta(x_j) c_k(x_j) - C_k)\big|_{\theta = \theta^{(t)}}$.
    Aggregate penalty gradients $\nabla_\theta \hat{Q} = \frac{2}{B(B-1)} \sum_{i=1}^B \sum_{j=i+1}^B \sum_{k=1}^m \nabla_\theta \hat{Q}_{ijk}$.
    Compute update $\theta^{(t+1)} = \theta^{(t)} - \eta(\nabla_\theta \hat{L} + \lambda \nabla_\theta \hat{Q})$.
**end for**
Return $d_{\theta^{(N)}}$. =0

---

For a fixed penalty coefficient $\lambda$, our algorithm performs stochastic gradient descent by sampling batches. Each batch is used to compute an unbiased estimate of the gradient of the loss $\ell(d_\theta)$ as well as an unbiased estimate of the gradient of the second-order penalty. The optimal choice of $\lambda$ is determined by standard hyperparameter tuning.

## 4 RELATED WORK

Our work builds on the constrained optimization view of fairness in machine learning. This view was first introduced in Zafar et al. (2015) and later extended in Goh et al. (2016). These works have focused on the convex setting, where optimality and convergence can be guaranteed. Although less is known in the non-convex case, there is work which frames the constrained optimization problem as a two-player game (Cotter et al., 2018b). The resulting classifier in this case is a randomized distribution over classifiers.

In contrast, our work proposes the use of second-order penalties in a general setting. This allows one to avoid the two-player game formulation in the non-convex case and accordingly the resulting classifier is deterministic. We are not the first to propose training for fairness in this manner: Donini et al. (2018) studies this for kernel methods and Komiyama et al. (2018) gives results for training a linear model with such penalties. In contrast, our methods are applicable to highly non-convex models and previous works do not address how to optimize for these penalities stochastically.

Another approach for non-convex settings is the use of adversarial training (Edwards & Storkey, 2015). In this setting, a predictive model is trained concurrently with an adversary, whose objective is to recover sensitive or protected attributes from the model outputs. The model's loss is then augmented with a penalty based on the adversary's success. This is thus another form of a two-player game, and hence also suffers from convergence issues. Our approach avoids these issues by allowing the use of a fixed penalty coefficient in training. Moreover, the form of our constraints may be seen as equivalent to an adversarial formulation in which the adversary is parameterized by a linear model.

The quadratic penalties we propose are similar to previous notions of orthogonality in machine learning, which is generally useful when one desires diversity of features (Xie et al., 2015; Xie, 2015; Kulesza et al., 2012). The specific penalties we impose may be interpreted as penalizing the Frobenius norm of the Gram matrix of model outputs, as mentioned in Xie et al. (2018). Many of these methods propose optimization schemes which are not amenable to stochastic mini-batching. In contrast, one of the key contributions of our work is showing that a second-order penalty may be optimized stochastically. Our result hinges on a standard calculus identity relating a product of integrals to a double integral. Similar techniques have been used previously in the context of reinforcement learning. Specifically, double-sampling is a known technique for unbiased minimization of the Bellman error, also a square of an expectation (Antos et al., 2008).

## 5   SIMULATIONS: LEARNING ORTHOGONAL REPRESENTATIONS

### 5.1   VISUALIZATION OF IRIS DATASET

We use the Iris dataset (Lichman et al., 2013) and train a simple model which is a network with a single intermediate 2-node layer.We show the effects of adding a penalty to encourage orthogonality on this layer. That is, we wish for the two learned features to be decorrelated over the dataset. We show the results in Figure 1 as weight on the penalty increases. We further increase the difficulty task by using small stochastic batches of size 4. While this is only a toy example, it clearly shows the possibility of stochastically training with these second-order penalties.

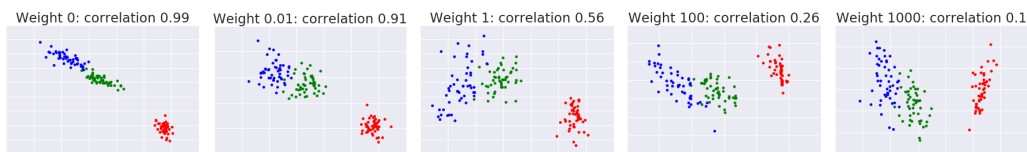

Figure 1: The learned 2d representation of the datapoints (colored by label) as the penalty weight changes. We see that indeed, the two features become decorrelated as we increase the weight.

### 5.2   CONVOLUTIONAL AUTOENCODER ON MNIST

We now move to applying our technique to a highly non-convex, neural network model. We use a convolutional autoencoder applied to MNIST images, where the encoder consists of 3 convolutional and max-pooling layers and the decoder consists of 4 convolutional and 3 upsampling layers. The encoded representation is 128-dimensional. We train this autoencoder with added penalties that encourage *orthonormality*. That is, in additional to orthogonality (as in the previous simulation), we also encourage the feature columns to be of unit norm. For training, we trained with the Adam optimizer and batch-size 128. We show in Figure 2 that indeed, we can stochastically optimize with the quadratic penalty and that feature correlation on the test set decreases as we increase the penalty coefficient. We see that we can dramatically decrease the correlation of features while suffering small sacrifices in reconstruction error.

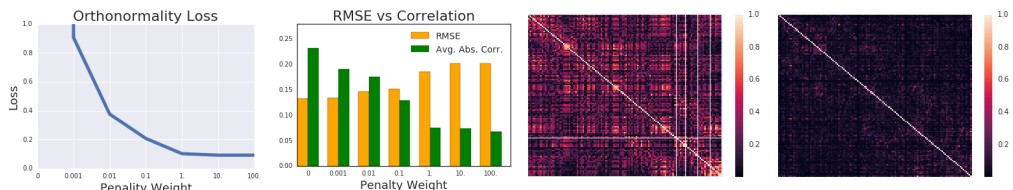

Figure 2: MNIST Autoencoder. **First**: Orthonormality loss on testing set as we change the penalty weight. **Second**: RMSE reconstruction error and average absolute pairwise feature correlation on the testing set across different penalty weight setttings. **Third**: Correlation heat map trained with no penalty. **Fourth**: Correlation heat map trained with penalty weight 1. We show the absolute values of the correlations.

## 6   EXPERIMENTS: FAIRNESS

### 6.1   DATASETS

**Adult** (Lichman et al., 2013) (48842 examples). Each datapoint corresponds to an individual and the task is to predict whether the person's income is more than 50k per year. We use 2 protected groups based on gender and preprocess the dataset and use a linear model, consistent with previous works, e.g. Zafar et al. (2015); Goh et al. (2016). The 2 fairness constraints here are the equal opportunity constraints for the 2 protected classes with slack 0.05, that is, the positive prediction rate on the positively-labeled examples for each protected class must be at least 95% of the overall positive prediction rate over all positively-labeled examples.

**Bank Marketing** (Lichman et al., 2013) (45211 examples). The data is based on a direct marketing campaign of a banking institution. The task is to predict whether someone will subscribe to a bank product. We use age as a protected feature and we have 5 protected groups, based on age quantiles and the 5 fairness constraints are demographic parity.

**Communities and Crime** (Lichman et al., 2013) (1994 examples). Each datapoint represents a community and the task is to predict whether a community has high (above the 70-th percentile) or low crime rate. We preprocess the data and use a linear model, consistent with previous works, e.g. Cotter et al. (2018a) and form the protected group based on race in the same way as done in Cotter et al. (2018a). We use four race features as real-valued protected attributes corresponding to White, Black, Asian and Hispanic. We threshold each at the median to form 8 protected groups. There is one fairness constraint for each of the 8 protected groups, which constrains the groups false positive rate to be at most the overall false positive rate.

**ProPublicas COMPAS** recidivism data (7, 918 examples). The task is to predict recidivism based on criminal history, jail and prison time, demographics, and risk scores. We preprocess this dataset in a similar way as the Adult dataset and the protected groups are two race-based (Black or White) and two gender-based (Male or Female). We use a 2-layer Neural Network with ReLU activations and 10 hidden units. The 4 fairness constraints here are the equal opportunity constraints for the 4 protected classes each being bounded by at most 0.05. That is, we wish to not predict recidivism more than 5% on top of the overall predicted recidivism rate restricted to examples whose which indeed had recidivism in two years for any protected class.

## 6.2 BASELINES

**Deterministic Lagrangian**: This method (Goh et al., 2016) is jointly training the Lagrangian in both model parameters and Lagrange multipliers and uses a Hinge approximation of the constraints to make the Lagrangian differentiable in its input. We then return the "best" iterate selected using a heuristic introduced by Cotter et al. (2018b), which finds a reasonable accuracy/fairness trade-off.

**Stochastic Lagrangian**: This method (Cotter et al., 2018b) returns a stochastic solution to the two-player game with the Lagrangian of the previous method as pay-off function. This solution is based on approximating a Nash equilibrium to this two-player game.

## 6.3 HYPERPARAMETER TUNING

We optimize over hyperparameters for our method in the following way: we perform a grid search over the weight of the orthogonality penalty as well as the fixed learning rate for the model trained using Adam optimizer. Then, to choose the best model, we find the highest accuracy model on the validation set which satisfies the constraints on the validation set. A final evaluation is performed on a fully un-seen test set. The Lagrangian baselines take in the desired slack and the hyperparameter search is over the fixed learning rate and is chosen using the heuristic on a validation set which chooses the best accuracy and constraint trade-off as done in Cotter et al. (2018b).

Table 1: Adult Experiment Results

| Algorithm | Train Error | Test Error | Train Violation | Test Violation |
|---|---|---|---|---|
| Unconstrained | 14.32% | 14.32% | 0.0688 | 0.0445 |
| Stochastic Lagrangian | 14.78% | 14.92% | 0 | 0 |
| Deterministic Lagrangian | 14.30% | 14.52% | 0 | 0 |
| Our Method | 14.31% | 14.32% | 0.0025 | 0.0092 |

We see that our method attains far lower constraint violation compared to training without constraints with almost no trade-off in accuracy. The Lagrangian baselines give solutions that satisfy the constraints but with considerable trade-off in accuracy.

Table 2: Bank Marketing Experiment Results

| Algorithm | Train Error | Test Error | Train Violation | Test Violation |
|---|---|---|---|---|
| Unconstrained | 9.48% | 9.37% | 0.0202 | 0.0152 |
| Deterministic Lagrangian | 9.64% | 9.55% | 0 | 0 |
| Stochastic Lagrangian | 9.39% | 9.51% | 0 | 0 |
| Our Method | 9.32% | 9.44% | 0 | 0 |

We see that our method attains the best testing error compared to the baselines while satisfying the fairness constraints.

Table 3: Communities and Crime Experiment Results

| Algorithm | Train Err. | Train Vio. | Train FPR | Test Err. | Test Vio. | Test FPR |
|---|---|---|---|---|---|---|
| Unconstrained | 9.93% | 0.1802 | 0.0518 | 15.43% | 0.3482 | 0.1108 |
| Determ. Lagrangian | 32.70% | 0 | 0.0700 | 34.26% | 0.0153 | 0.0554 |
| Stoch. Lagrangian | 10.34% | 0 | 0.0238 | 15.29% | 0.1129 | 0.0576 |
| Our Method | 14.24% | 0.0948 | 0.0252 | 17.43% | 0.0903 | 0.0408 |

We show both the violations and overall false positive rates (FPR). The violation is the maximum difference between the FPR for any given protected group and the overall FPR. For example, in the first row, we see that there exists a protected group with a FPR of $0.3482 + 0.1108 = 0.459$ by the model. While our method does not attain the lowest violation, it provides a reasonable trade-off. Interestingly, our method attains the lowest *overall* (i.e. *average*) FPR across the entire dataset out of all the methods. A low violation at the cost of high overall FPR may be undesirable because ensuring fairness in FPR might make everyone worse off in terms of FPR.

Table 4: COMPAS Experiment Results

| Algorithm | Train Error | Test Error | Train Violation | Test Violation |
|---|---|---|---|---|
| Unconstrained | 30.56% | 31.09% | 0.1151 | 0.1082 |
| Deterministic Lagrangian | 28.40% | 32.23% | 0.0803 | 0.0800 |
| Stochastic Lagrangian | 37.11% | 36.76% | 0 | 0.0284 |
| Our Method | 35.39% | 33.68% | 0 | 0.0062 |

We see that our method is able to learn a classifier that is significantly closer to satisfying the fairness constraints than the baselines while trading off a reasonable amount of accuracy.

## 7    CONCLUSION

We have presented a method for stochastically learning with second-order penalties. Such penalties may be used in a number of applications. We have shown how they can be used to encourage the learning of orthonormal features. We have additionally demonstrated their applicability to fairness, where they provide a more stable training procedure while yielding at least competitive final performance.

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

## A    PARETO CURVES

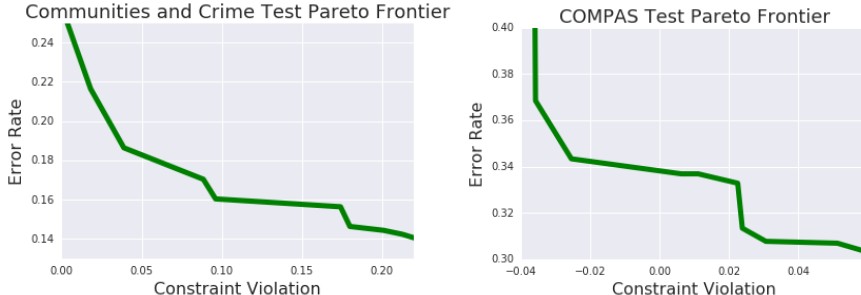

Figure 3: For two of our datasets, we show the pareto frontiers for the testing error vs constraint violation trade-off over the runs of our method obtained from the grid search on the single constant learning rate and the orthogonality penalty term.

