# OpenReview forum: "Stochastic Learning of Additive Second-Order Penalties with  Applications to Fairness"
_ICLR.cc/2019/Conference_

### Official Review · AnonReviewer3 · 2018-11-02
**Unclear contribution**

**Rating:** 4
**Confidence:** 3

**Review:**

The authors propose a method to generate predictions under fairness constraints. The main idea is to take linear fairness constraints, and replace them with weak squared penalties plugged into the objectives, which supposed to help in cases where the loss function is not convex. The penalty coefficients are chosen by cross-validation, and the effectiveness of this approach is demonstrated empirically.

In Sec. 3.1, the authors point out several shortcomings of using linear penalties (using Lagrange multipliers) for non-convex losses. These seem valid. Sec. 3.2, however, is not clear on why exactly replacing the linear penalties with quadratic penalties solves these issues. I'm hoping the authors can clarify the following points:

1) The authors note that, for quadratic penalties, \lambda->0 means no constraints, and \lambda->\infty means hard constraints. Isn't this also true for linear constraints?

2a) Why do linear penalties have unique \lambda_k for each constraint k, but the quadratic objective has only a single \lambda for all constraints?

2b) Why can CV over \lambda be used for quadratic constraints - what is the justification? And, more importantly, why *can't* it be used with linear constraints? If it can, then this should be one of the baselines compared to in the experiments.

3) What is the criterion optimized for by CV - accuracy or the constraints? Different parts of the paper give different answers to this question. For example, "... may be easily determined via standard hyperparameter optimization methods" vs. "tuning \lambda to directly satisfy the desired fairness metrics". Or even more unclear - "choose \lambda ... so that the final solution gives the desired fairness-accuracy trade-off". How is the desired trade-off defined?

4) If there is a trade-off between fairness and accuracy, and no clear-cut criterion for evaluation is pre-defined, then the evaluation procedure should compare methods across this trade-off (similarly to precision-recall analysis).

5) The authors differentiate between cases where the loss is either convex or non-convex. This is confusing - most losses are convex, and non-linearity appears when they are composed with non-linear predictors. Is this the case here? If so, the fairness constraints are no longer linear, and they're quadratic counterpart is no longer quadratic. It would be helpful if the authors specify where the non-linearity comes from, and what they assume about the loss and predictors.

6) Why is it important to show that the quadratic constraints can be written as an expectation? Isn't the square of an expectation always an expectation of pairs? How does the double summation/integral effect runtime?

7) It would be helpful if the authors differentiate between loss/constraints over the entire distribution vs. over a given sample set.

---

> ### Author Response · Authors · 2018-11-21
> **Response**
>
> We thank the reviewer for their feedback.  Our responses to the questions are below:
>
> 1. In the case of linear penalties, the lambdas are coefficients on the constraints <d, c>.  While lambda=0 corresponds to an unconstrained loss, lambda->infty does not correspond to a hard constraint.  Instead, it corresponds to a loss which encourages <d, c> to go to negative infinity.  This does not correspond to the desired fair classifier, which should have <d, c> = C, a finite constant.
>
>
> 2a. Note that the Lagrange multipliers in the first equation in Sec 3.1 appear on both positive and negative forms of the constraints.  This means that for an arbitrary setting of Lagrange multipliers, some constraints will have a positive coefficient and others will have a negative coefficient.  Accordingly, the coefficients cannot be combined to one.  In contrast, in the second-order penalty form, all constraints must be associated with a positive coefficient, and thus for ease of training we may simplify training by using a single lambda.
>
> 2b. Because linear penalties in general require multiple Lagrange multipliers and that a deterministic equilibrium is not guaranteed, a direct application of CV may be computationally expensive.  We are not aware of previous work which proposes to optimize the Lagrangian dual this way.
>
> 3. In practice, we considered solutions that satisfied the constraints on the validation set and chose the one with lowest error. If there were no solutions satisfying the constraints, then we chose the setting which had the lowest fairness violation on the validation set.
>
> 4. We use the method just mentioned to choose the fairness/accuracy trade-off. While the pareto curves may have been compared, we note that these solutions may not necessarily be attained in practice and that in practice, one must commit to one hyperparameter setting based on training/validation data.
>
> 5. We assume the loss and constraints are defined with respect to classifier outputs, while the classifier output is defined with respect to model parameters.  While the loss and second-order constraints with respect to classifier outputs is convex, the same quantities with respect to model parameters in general can be non-convex.
>
> 6. The derivations from Eq. 4 to Eq. 7 are provided to show how second-order penalties may be optimized in an SGD setting.  That is, directly optimizing Eq 4 would require a full-batch gradient (very slow and in some cases intractable), while optimizing Eq 7 only requires successive mini-batch gradients (very fast and scalable).
>
> 7. In machine learning fairness, both losses and constraints are typically defined in terms of expectations over the entire dataset.  That is, the loss is an expectation of example-wise loss (as is standard in machine learning).  The constraints are a weighted expectation of classifier predictions over the entire dataset (as elaborated in Sec 2).

---

### Official Review · AnonReviewer2 · 2018-11-05
**Experiments and motivation can be improved**

**Rating:** 5
**Confidence:** 4

**Review:**

In this paper, the authors propose a method for optimizing quadratic penalties over non-convex loss functions. The authors motivate their approach by showing the complexity of training non-convex models with linear constraints and proposing a simpler method to introduce these constraints as regularization terms. Finally, they show how their proposed method compares to alternative solutions on a series of benchmarks.

Use of double sampling method for estimating the loss and second order penalty appears to be novel, however, there is no discussion of the implications of using this method to train non-convex models --- one would suspect that use of double sampling may make the gradient descent susceptible to high variance. Simulation results in the paper only demonstrate how the use of the loss changes the solution but there is no discussion or experiments on complexity of training models that use this approach. This is particularly important because authors are claiming their method does not suffer from complexity issues which other methods suffer, but this claim is not supported by any evidence. For example, how many iterations were necessary to train the model? How sensitive is the training to initial conditions and changes in hyper parameters?

Authors motivate their method by pointing out the limitations of using fairness constraints in non-convex models, however, they don’t provide-sufficient evidence for why non-convex models are actually useful in their experiments --- the datasets they used are small and models that are actually convex may perform just as good or nearly as good as highly non-linear, non-convex models which authors are trying to use.

Overall, I would recommend the authors to improve the presentation by providing more context for the use of double sampling method and other relevant works in this area (at least showing the impact of using double sampling on training). Moreover, given the datasets and scope of questions related to fairness they need to provide better experiments that motivate the use of their method (compared to simpler methods) or consider other problems where their approach could be more useful. Given these considerations I believe this paper does not meet the standards for publication.

---

> ### Author Response · Authors · 2018-11-21
> **Response**
>
> We thank the reviewer for their feedback.
>
> 1. Because of the Lagrangian duality gap in the non-convex setting, the Lagrangian might not even have a saddle point to converge to and thus training both the parameters and multipliers jointly might never converge. With our method, since there is a fixed penalty weight, then there is at least some guarantee that we will converge to a minima or stationary point.
>
> 2. We used four different datasets for the fairness experiments, which are the standard publicly available benchmarks in the field of machine learning fairness (e.g. see [1,2,3]).  While some of the experiments use convex models, several are performed on non-convex neural networks.  We agree that these fairness datasets and models may be on the small side which is why we also included the simulation experiment on MNIST, which uses a convolutional neural network.
>
> 3. We agree that the presentation and context could be clarified further. We emphasize that the double sampling method allows us to effectively train with orthogonality penalties, an approach that is very relevant to training fair classifiers, but has only been done for linear models [4] and kernel methods [5]. Our method is the first which allows us to do so for general methods.
>
>
>
> [1] Zafar, Muhammad Bilal, et al. "Fairness Constraints: Mechanisms for Fair Classification." AISTATS 2017.
> [2] Goh, Gabriel, et al. "Satisfying real-world goals with dataset constraints." NIPS. 2016.
> [3] Agarwal, Alekh, et al. "A reductions approach to fair classification." ICML 2018
> [4] Komiyama, Junpei, et al. “Nonconvex optimization for regression with fairness constraints.” ICML 2018.
> [5] Donini, Michele, et al. "Empirical Risk Minimization under Fairness Constraints." NIPS 2018.

---

### Official Review · AnonReviewer1 · 2018-11-06
**Motivation and contribution are not very clear**

**Rating:** 5
**Confidence:** 3

**Review:**

In this paper, the authors propose a method to generate predictions under fairness constraints by optimizing quadratic penalties over non-convex loss functions. The main idea is to replace the linear fairness constraints by the second-order penalty. Meanwhile, an efficient method is derived to compute the gradients associated with the second-order penalties in stochastic mini-batch settings. Finally, the experimental results show the effectiveness of their proposed method empirically.

(1) The authors argued that their experimental results in a more practical training procedure in non-convex, large-data settings. However, in Section 6, the sample size of the data-sets they used are small and the loss functions of learning models are convex. I think they need to provide more experimental results to make their proposed method more convincing.

(2) It is a little difficult to follow the motivation and contributions of this paper. I would recommend the authors to improve the presentation by providing more context for the use of double integral or sampling method and other mostly relevant works in this area.

(3) From the optimization viewpoint, the second-order penalty in Eq. (3) is convex with respect to d. Why replacing the linear penalties with quadratic penalties to solve the shortcomings of using linear penalties or Lagrange multipliers. Isn't the square of an expectation always an expectation of pairs? In other word, Eq. (4) is always equivalient to Eq. (7) without additional conditions?

---

> ### Author Response · Authors · 2018-11-21
> **Response**
>
> We thank the reviewer for their feedback.  Our responses to the three main points are below:
>
> 1. We used four different datasets for the fairness experiments, which are standard benchmarks in the field of machine learning fairness.  We note that well-known previous works on fair classification e.g. [1,2,3] evaluate on only a subset of these datasets: [1] uses Bank and Adult, [2] uses Adult, and [3] uses Adult and COMPAS, although [2,3] also have some additional experiments but on datasets which are not publicly available. Moreover, on all of these datasets, we train with more constraints/protected groups simultaneously thus making the task more difficult. For example, in [1] and [3], each of the datasets had only 1 protected group while in our work for these datasets, we had 4 protected groups.
>
> Furthermore, while some of the experiments use convex models, several are performed on non-convex neural networks (both simulations and the COMPAS dataset).
>
> 2. The reviewer is right that we should clarify better the contributions and how they relate to previous work. The main contribution is as follows: previous works have approached fair classification via orthogonality on linear models [4] and kernel methods [5], but we provide a method that works in general to any parametric model that can be trained using SGD.
>
> 3. The main issue of linear penalties is that in non-convex settings (e.g. when using a multi-layer neural network), the Lagrangian dual is difficult to optimize and there may not even exist an appropriate saddle point to converge to.  With this in mind, the main contribution of our paper is to introduce second-order penalties, which are useful due to the fact that any fixed choice of lambda yields a deterministic solution, and the solutions derived according to lambda=0 to lambda->infty can provide a spectrum of fairness-accuracy trade-offs.  The derivations from Eq. 4 to Eq. 7 are then provided to show how second-order penalties may be optimized in an SGD setting.  That is, directly optimizing Eq 4 would require a full-batch gradient, while optimizing Eq 7 only requires successive mini-batch gradients.
>
> [1] Zafar, Muhammad Bilal, et al. "Fairness Constraints: Mechanisms for Fair Classification." AISTATS 2017.
> [2] Goh, Gabriel, et al. "Satisfying real-world goals with dataset constraints." NIPS. 2016.
> [3] Agarwal, Alekh, et al. "A reductions approach to fair classification." ICML 2018
> [4] Komiyama, Junpei, et al. “Nonconvex optimization for regression with fairness constraints.” ICML 2018.
> [5] Donini, Michele, et al. "Empirical Risk Minimization under Fairness Constraints." NIPS 2018.

---

### Meta-Review · Area_Chair1 · 2018-12-13

**Confidence:** 5
**Recommendation:** Reject

**Metareview:**

The paper presents a method to stochastically optimize second-order penalties and show how this could be applied to training fairness-aware classifiers, where the linear penalties associated with common fairness criteria are expressed as the second order penalties.

While the reviewers acknowledged the potential usefulness of the proposed approach, all of them agreed that the paper requires: (1) major improvement in clarifying important points related to the approach (see R3’s detailed comments; R2’s concern on using the double sampling method to train non-convex models; see R1’s and R3’s concerns regarding the double summation/integral terms and how this effects runtime), and (2) major improvement in justifying its application to fairness; as noted by R2, “there is no sufficient evidence why non-convex models are actually useful in the experiments”. Given that fairness problems are currently studied on the small scale datasets (which is not this paper’s fault), a comparison to simpler methods for fairness or other applications could substantially strengthen the contribution and evaluation of this work.
We hope the reviews are useful for improving and revising the paper.